# Waterlogging Causes Early Modification in the Physiological Performance, Carotenoids, Chlorophylls, Proline, and Soluble Sugars of Cucumber Plants

**DOI:** 10.3390/plants8060160

**Published:** 2019-06-08

**Authors:** T. Casey Barickman, Catherine R. Simpson, Carl E. Sams

**Affiliations:** 1Department of Plant and Soil Sciences, Mississippi State University, North Mississippi Research and Extension Center, Verona, MS 38879, USA; 2Department of Agriculture, Agribusiness, and Environmental Sciences, Texas A&M University-Kingsville, Kingsville, TX 78363, USA; catherine.simpson@tamuk.edu; 3Department of Plant Sciences, University of Tennessee, Knoxville, TN 37996, USA; carlsams@utk.edu

**Keywords:** anoxia, photosynthesis, stomatal conductance, sucrose, proline

## Abstract

Waterlogging occurs because of poor soil drainage and/or excessive rainfall and is a serious abiotic stress affecting plant growth because of declining oxygen supplied to submerged tissues. Although cucumber (*Cucumis sativus* L.) is sensitive to waterlogging, its ability to generate adventitious roots facilitates gas diffusion and increases plant survival when oxygen concentrations are low. To understand the physiological responses to waterlogging, a 10-day waterlogging experiment was conducted. The objective of this study was to measure the photosynthetic and key metabolites of cucumber plants under waterlogging conditions for 10 days. Plants were also harvested at the end of 10 days and analyzed for plant height (ht), leaf number and area, fresh mass (FM), dry mass (DM), chlorophyll (Chl), carotenoid (CAR), proline, and soluble sugars. Results indicated that cucumber plants subjected to the 10-day waterlogging stress conditions were stunted, had fewer leaves, and decreased leaf area, FM, and DM. There were differences in physiological performance, Chl, CAR, proline, and soluble sugars. Overall, waterlogging stress decreased net photosynthesis (*A*), having a negative effect on biomass accumulation. However, these decreases were also dependent on other factors, such as plant size, morphology, and water use efficiency (WUE) that played a role in the overall metabolism of the plant.

## 1. Introduction

Agricultural crops can be subjected to soil waterlogging, a major abiotic stress, caused by excessive precipitation and poor soil drainage, which can dramatically reduce the plant’s physiological performance, yield, and fruit quality. An early response to waterlogging, especially in sensitive crops species such as cucumber, is the reduction of water uptake in the root system [1,2] Decreasing root hydraulic conductance, the reduction of water uptake, is a result of a disruption of aquaporin function [3]. Additionally, waterlogging causes the production of the high-energy phosphate compound, ATP, to decrease up to 37.5% in affected plant cells [4]. Previous research has demonstrated that net photosynthesis (*A*) rate declined rapidly in cucumber plants after two days of root-zone hypoxia treatment [5]. Thus, the repercussions of the plants declining energy reserves due to the decreased *A* of waterlogged plants are metabolically widespread, even for a short period of time. Tolerance to waterlogging, especially in cucumber, is induced by the initiation and proliferation of adventitious roots. To minimize the distance for oxygen diffusion and improve gas diffusivity, cucumber plants form adventitious root as a key adaptation to waterlogging [6].

Molecular, metabolic, and physiological traits of plants with a tolerance to waterlogging are related to how the plants express these changes under strong environmental influences [7,8]. These types of responses result in decreases in cellular energy, cytoplasmic pH, stem elongation, production of adventitious roots, and production and accumulation of toxic metabolites and reactive oxygen species [9,10]. Additionally, fresh and dry mass (FM; DM) accumulation is significantly decreased with waterlogging events [11]. However, tolerance to waterlogging is indicative of many of these molecular, metabolic, and physiological responses. For instance, He et al. [12] demonstrated that waterlogged cucumber plants during the fruiting stage had decreases in biomass accumulation and gas exchange parameters such as *A* and stomatal conductance (g_s_). Furthermore, waterlogging stress might impair the photosynthetic electron–transport chain, limiting the rate of CO_2_ assimilation or *A* during fruit set of cucumber. Most of the research on cucumber waterlogging has focused on the effects of the root system and how the expression of genes profiles is affected. For example, Qi et al. [13] analyzed differentially expressed genes from carbon metabolism, photosynthesis, reactive oxygen species, and hormone synthesis and signaling was mostly downregulated when plants were subjected to waterlogging conditions for 24 h. On the other hand, previous research has also indicated that waterlogging cucumber plants for 48 h increased enzymes, such as pyruvate decarboxylase, alcohol dehydrogenase, and lactate dehydrogenase, associated with anaerobic fermentation in root tissue [14]. 

However, there has been a lack of information on cucumber seedlings that are associated with early growth and development, and physiological performance. Thus, the purpose of the current study is to demonstrate the physiological effects, such as *A*, g_s_, internal CO_2_ (C_i_) concentrations, transpiration (*E*), and water use efficiency (WUE) of a 10-day waterlogging event on the early stages of growth in cucumber plants. The objectives of the study are to (1) determine how early growth stage waterlogging affects cucumber biomass accumulation, morphological attributes, carotenoids (CAR), chlorophylls (Chl), proline, and soluble sugars; and (2) determine the effects of early growth stage waterlogging on *A*, g_s_, C_i_, *E*, and WUE of cucumber plants. 

## 2. Results and Discussion

In plant photosynthetic reactions, molecular oxygen is involved as a catalyst to facilitate the assembly of energy compounds leading to the production of glucose. Under a variety of stress conditions, plants reduce photosynthetic efficiencies, growth, and development, and yields as an avoidance strategy. Thus, examining the response of cucumber seedlings to waterlogging stress early in the developmental stage may help to understand the mechanisms of tolerance to these conditions. During the study, waterlogging of cucumber plants over 10 days induced several physiological changes, including reduction of growth, FM and DM, and photosynthesis when compared to the non-waterlogged treated cucumber plants. There have been similar studies on other cucurbits such as cucumber [5], summer squash [15], and watermelon [16]. 

Previous research by He et al. [12] found that there were 18% and 49% reductions of plant height and fresh weight cucumber compared the non-waterlogged, respectively. In the current study, there were similar results. Total plant height was reduced by waterlogging stress (*p* = 0.05) after a 10-day treatment (Table 1). Waterlogged plants had a 12.3% decrease in plant height ompared to the non-waterlogged plants. 

The number of leaves (*p* = 0.05) and leaf area (*p* ≤ 0.0001) were also significantly decreased by the 10-day waterlogging treatment (Table 1). On average, leaf number and leaf area (Table 1) were reduced under the 10-day waterlogging treatment by 11.0% and 36.8%, respectively. Results demonstrate that cucumber plant FM (*p* ≤ 0.0001), DM (*p* ≤ 0.001), and the ratio of DM:FM (*p* ≤ 0.05) were significantly affected by the 10-day waterlogging treatment (Table 1). Cucumber plant FM and DM was reduced by 35.2% and 26.0%, respectively, when compared to the non-waterlogged treatment. There was an increase in the DM:FM ratio of cucumber plants subjected to the 10-day waterlogging treatment with a 12.5% difference relative to the non-waterlogged treated plants. The results demonstrate that leaf area, FM, and DM were the most affected by the 10-day waterlogging. Overall, there was a similar trend of decreasing the morphological attributes and biomass. Furthermore, the decrease in DM could be a direct symptom of the reduction in leaf area on waterlogged cucumber plants. Huang et al. [15] also established that waterlogging summer squash for 14 days led to a reduction in root and shoot growth, and fruit yield. Additionally, waterlogging increased the production of adventitious roots. Both observations were consistent with the current study where waterlogging cucumber plants for 10 days decreased leaf number, FM, DM, and plant heights. Data also revealed that there was a decrease in percent water content in the waterlogged treated cucumber plants (Table 1). The decrease in water content of waterlogged plants can be associated with the reduction in other morphological changes such as plant height, leaf area, and biomass accumulation. There were also increases in adventitious roots (data not shown) in the 10-day waterlogged treated cucumber plants compared to the non-waterlogged plants. Previous research on wheat indicated that waterlogging wheat, even for only three days, had a long-lasting effect on growth. For example, plant height, shoot mass, and seminal root mass was two to three times lower in waterlogged plants compared to the non-waterlogged control plants [17]. 

Available research information indicates that agricultural plants subjected to waterlogging have decreased photosynthetic capacity. In the current study, there were significant interactions between *A*, C_i_, g_s_, and *E* in cucumber plants subjected to water stress (Figure 1 A–D). This is supported by findings from other similar studies. For example, tomato plants flooded within the first hour had a reduction in leaf water potential, g_s_, and *E* [18]. In another study, waterlogging of tomato plants significantly decreased *A*, C_i_, g_s_, and *E* over six days [19]. In the current study, the effect of waterlogging on *A* in cucumber plants were significantly (*p* ≤ 0.0001) reduced over the 10-day treatment period compared to the non-waterlogging treatment (Figure 1A). 

The waterlogging treatment reduced the average *A* across the 10-day treatment period by 63.3% compared to the non-waterlogging treatment. Cucumber plants had a negative response to the 10-day waterlogging treatment with an *A* of 19.23 µmol CO_2_·m^−2^·s^−1^ one day after treatment (DAT) to 7.26 µmol CO_2_·m^−2^·s^−1^ nine DAT. There was also a significant positive response in *A* for cucumber plants in the non-waterlogging treatment with an *A* of 17.08 µmol CO_2_·m^−2^·s^−1^ one DAT to 19.80 µmol CO_2_·m^−2^·s^−1^ nine DAT. Consequently, there were no significant differences in *A* on one and two DAT. However, by the three DAT, there was a 31.4% reduction in *A* when comparing the waterlogged to the non-waterlogged cucumber plants. Results indicate that the trend continued until the end of the 10-day waterlogging treatment. Previous research indicated similar results from studies involving summer squash [15] and watermelon [20]. The assimilation of atmospheric CO_2_ is of central importance to plant metabolism, as plants chemically reduce carbon. These reactions represent the acquisition of stored chemical energy, which ultimately provides the carbon skeletons of the plants structure. Approximately 96% of the total dry mass is made up of carbon, hydrogen, and oxygen assimilated into organic molecules by photosynthesis [21]. In stress conditions, such as waterlogging, decreases in *A* can have a negative effect on biomass accumulation as demonstrated in the current study in which cucumber plants were subjected to the stress for 10 days. The decrease in *A* can be associated with the decrease in biomass when compared to the non-waterlogged plants. Even though there was a decrease of 63.3% in *A*, the decrease of biomass in waterlogged plants was approximately 26% giving the notion that other factors, such as plant size, morphology, and WUE play a role in the overall metabolism of the plant. The C_i_ concentrations of cucumber plants subjected to a 10-day waterlogging treatment had a significant (*p* ≤ 0.001) increase from one to nine DAT (Figure 1B). However, there was a significant decrease in C_i_ on two DAT, while there was a recovery at three to nine DAT. Previous research examining self-grafted and non-grafted tomato plants under waterlogging conditions found similar results in that C_i_ increased 3 to 23% in both types of tomato plants when compared to the same types of plants under non-waterlogged conditions [19]. In the current study, there was a positive response to C_i_ in waterlogged treated plants with a C_i_ ranging from 360.23 µmol·m^−2^·s^−1^ at two DAT to 382.22 µmol·m^−2^·s^−1^ at six DAT. Non-waterlogged cucumber plants saw a similar trend in C_i_. Non-waterlogged plants had a low of 350.56 µmol·m^−2^·s^−1^ at two DAT compared to a C_i_ of 365.85 µmol·m^−2^·s^−1^ at one DAT. However, there were significant decreases in waterlogged plant C_i_ when comparing non-waterlogged-treated cucumber plants. The stress response of cucumber plants subjected to waterlogging was not as pronounced for g_s_ and *E* compared to the non-waterlogged plants (Figure 1C,D). Previous research has demonstrated that flooding stress for six days significantly reduced *E* in watermelons [16] and cucumbers subjected to nine-day hypoxia stress [12]. However, in the current study, cucumber plants in both waterlogged and non-waterlogged treatments demonstrated on two and three DAT, there were significant (*p* ≤ 0.001) differences between the two treatments. The decrease in C_i_, g_s_, and *E* in both waterlogged and non-waterlogged cucumber plants may have been the cause of transplanting the seedlings from the germination trays to the pots.

The combined regression analysis demonstrated a strong quadratic relationship between g_s_ to cucumber plant *E* (Figure 2A), C_i_ (Figure 2B), and WUE (Figure 2C) in cucumber plants. There were strong relationships in both waterlogged and non-waterlogged plants that indicated increasing transpiration rates *E* with increased g_s_. However, there were no differences in the relationship between g_s_ and *E* between waterlogged and non-waterlogged cucumber plants.

The results also indicated a strong positive quadratic trend in cucumber plants for C_i_ with increasing g_s_. There were significant (*p* ≤ 0.0001) differences between waterlogged and non-waterlogged treated plants. Stomatal conductance is a common indication of plant–water status and plant–water balance. In water stress conditions, root hydraulic conductivity typically decreases and results in stomatal closure to reduce water loss. However, this does not seem to be the pattern reflected in this study. While *E*, C_i_, and g_s_ temporarily decreased after two DAT, they recovered and increased from three to nine DAT. Since g_s_ is responsive to almost all external and internal factors related to stress factors, it represents a highly integrative basis for the overall effect of waterlogging on photosynthetic parameters [9,22]. Previous research on g_s_ reactions could be due to hypoxia stress-induced responses related to ethylene production which possibly caused stomatal closure [23,24,25]. The WUE, estimated by calculating *A*/g_s_, showed a decreasing quadratic trend in both waterlogged and non-waterlogged treated plants as g_s_ increased and peaked roughly around the g_s_ value of 0.37 mol·m^−2^·s^−1^. There were also significant (*p* ≤ 0.0001) differences in WUE when comparing waterlogged to non-waterlogged treated cucumber plants. The decreased WUE of waterlogged cucumber plants can be explained by comparing early stress responses and their reduced ability to take up water compared to well-drained plants. For example, Else et al. [18] found that tomato plants that were flooded within the first hour of a 12 h photoperiod, had a significant decrease in leaf water potential from −0.55 MPa to approximately −0.80 MPa. Previous research has also indicated a decrease of WUE in cucumber subjected to waterlogging stress [26]. Additionally, the trend of decreased WUE can also be observed in the decreased amount of water content in the waterlogged treated cucumber plants (Table 1). In the current study, these results conflict with previous studies which found that waterlogging stress led to stomatal closure and reduced root hydraulic conductivities. However, these results indicate that WUE was reduced by other means, possibly because of downregulation of photosynthetic process and carbohydrate production while increasing water loss through stomatal opening and transpiration. 

Mechanisms, such as changes in chlorophyll (Chl) and carotenoid (CAR) concentration, non-radiative energy dissipation, and non-photochemical quenching [27,28], have evolved to help protect the plant from environmental stresses. CAR pigments protect the photosynthetic structures, such as photosystem I and II, by quenching excited triplet Chl to dissipate excess energy [29], and inhibiting oxidative damage by binding singlet oxygen [30,31]. Carotenoid concentrations are shaped by genetic, biochemical, and physiological attributes of a plant species, as well as environmental factors such as fertility, light, and temperature [32,33] or by stress factors such as drought [34] and flooding [35]. In the current study, waterlogging cucumber plants significantly changed the concentration of Chl and CAR compared to the non-waterlogged treatment (Figure 3). There were significant (*p* ≤ 0.05) decreases in Chl b in waterlogged treated compared to non-waterlogged treated plants. However, changes were small at an 8.8% difference in Chl b. Similarly, there was a 6.1% change in Chl a + b when comparing to the two treatments. Waterlogging resulted in visible yellowing of cucumber leaves, which caused decreases in Chl b concentrations. Other studies in mung bean [36], wheat [37], and onion [38] found that waterlogging decreases Chl and CAR concentrations. In the current study, cucumber plants developed a stress response to waterlogging, allowing them to increase their CAR concentrations when subjected to waterlogging for 10 days. Thus, there was a significant (*p* ≤ 0.05) increase of 6.0% in CAR concentrations in waterlogged compared to non-waterlogged plants (Figure 3). Under waterlogging, reduction of nitrogen and production of toxic substances were caused by photorespiration [39]. These results confirm our findings of increased levels of chlorosis over the time of the experiment. In the current study, the non-stomatal components limited *A*, which can be visualized in Figure 2B, where for a given g_s_, waterlogged plants showed higher C_i_ than non-waterlogged plant. This data reveals there was an association between a decrease *A* and lower leaf Chl concentrations in waterlogged treated cucumber plants. These results were similar to another study that associated reductions in photosynthetic pigment concentrations and photosynthesis [40].

Proline, an essential amino acid, is known to participate in the biosynthesis of primary metabolism during growth and development [41,42]. Proline also accumulates in response to the imposition of a wide range of stress responses in plants. Its role has focused on the ability of proline to mediate osmotic adjustment, stabilize subcellular structures, and scavenge free radicals in cases of plant stress [43,44]. In this study, the accumulation of proline as an osmolyte could have contributed to maintaining plant water status and hydraulic conductivity during waterlogging stress. Cucumber plants subjected to waterlogging stress for 10 days accumulated significantly higher amounts of proline compared to the non-waterlogged plants (Figure 4). More specifically, there was a 58.9% increase in proline concentrations in the leaf tissue of waterlogged cucumber plants. Previous research has demonstrated that barley plants increased proline concentrations after a 72- to 120-hours soil flooding experiment [45]. This could also emphasize the role of proline in membrane stabilization and maintaining cytosolic pH levels, along with other metabolic functions [46]. 

Induction of waterlogging stress causes an array of metabolic responses in which plants try to cope with an anaerobic environment. The plant’s ability to ferment available sugars for proper metabolic function, such as in rice [47], helps plants to tolerate a waterlogged environment. Previous research has demonstrated that there was a waterlogging-induced sugar metabolism in mung bean [36]. Results indicated that levels of total and non-reducing sugars declined in mung bean plants subjected to an eight-day waterlogging treatment. In the current study, there were differences in cucumber plant soluble sugar concentrations in the leaf tissue (Figure 5). There was a significant (*p* ≤ 0.001) decrease in sucrose by 39.5% from cucumber plants that were given a 10 day waterlogging treatment compared to non-waterlogged plants. Glucose levels were significantly (*p* ≤ 0.05) higher in waterlogged cucumber plants. However, there were no changes in fructose levels in the cucumber leaf tissue. The decreases in sucrose levels may be from the reduced level of *A* in waterlogged plants. The reduction of *A* causes an indirect reduction of sucrose in the plant. Waterlogged-induced stress has been linked to carbohydrate starvation [48]. Additionally, greater activity of sucrose synthase, a key enzyme responsible for the production of glucose and fructose from sucrose under oxygen deprivation, is associated with an increase in reducing sugars under waterlogging conditions [49]. Thus, the greater activity of sucrose synthase may explain why there was an increase in the availability of the reducing sugars in the cucumber plants subjected to waterlogging conditions. 

## 3. Materials and Methods 

The study was conducted from 16 November until 19 December 2018. Seeds of “Straight 8” cucumber (Johnny’s Selected Seed, Waterville, ME, USA) were sown into 10-cm pots filled with Pro-Mix BX soilless medium (Premier Tech Horticulture, QC, Canada) with a 15N-3.9P-9.9K controlled release fertilizer (Osmocote; Scotts Miracle-Gro, Marysville, OH, USA) incorporated at a rate of 5.93 kg·m^−3^. Seeds were germinated in a growth chamber (Model PGC-6L; Percival Scientific, Inc., Boone, IA) at 26/22 °C (day/night) temperature at 16 h photoperiod. Two weeks after emergence, all cucumber plants were placed in 11 L containers (Rubbermaid Inc., Wooster, OH, USA). Experimental treatments consisted of 1) waterlogging and 2) non-waterlogging for 10 days. One-half of the containers were filled with enough water to be 10 cm over the top of the pots to simulate waterlogging conditions for 10 days, and water was filled to the 3-cm level as needed. The other half of the plants were watered as needed. Light intensity averaged 350 µmol·m^−2^·s^−1^ over the entire photoperiod. The experimental design was a randomized complete block with four blocks and two replications within each block, where two containers represented a replication. There were six individual plants per container representing an experimental unit with one cucumber plant per pot. Thus, there were a total of 12 plants per treatment. 

*Physiological Performance and Plant Harvest.* One day after treatment (DAT) initiation and over the subsequent nine days, physiological performance measured with a Li-6800 (Li-Cor Biosciences, Lincoln, NE, USA) photosynthesis system were taken from the cucumber plants for *A*, g_s_, C_i_ concentrations, and *E*. Plants were harvested at the end of the 10 day experiment, and plant height, leaf number and area, FM, DM, CAR, Chl, proline, and soluble sugars were taken to calculate the horticultural performance of the cucumber plants. All leaves from each experimental unit were taken for leaf number and area. Plant heights were measured from the base of the plant to the meristem. All the above-ground cucumber plant tissue was then weighed for FM. Subsequently, the cucumber plant tissue was flash frozen in liquid nitrogen and plant in an ultra-low freezer at −80 °C. Cucumber leaf tissue was then freeze-dried (Model: FreeZone 2.5 L; LabConco Corp. Kansas City, MO, USA), weighed for DM, and then subsampled for total CAR, Chl, total proline, and soluble sugar concentrations of the cucumber leaf tissue. 

*Carotenoid and Chlorophyll Analysis.* Total carotenoid and chlorophyll pigments were extracted from freeze-dried tissues and analysis was performed according to methods described by Lichtenthaler and Buschmann [50]. Freeze-dried cucumber leaf tissues were weighed to 0.3 g and placed in a sterilized mortar. To neutralize plant acids, 100 mg of MgCO_3_ was added to the sample. Then 3 mL of diethyl ether was added and the mixture was homogenized to extract the pigments. The turbid pigment extract was transferred to a 15-mL centrifuge tube and the total volume was brought up to 5 mL with additional solvent. The samples were centrifuged for 5 min at 500 × *g* at 10 °C. The extract was then placed into a cuvette for spectrophotometric analysis. A Genesys 10S UV/Vis spectrophotometer (Thermo Fisher Scientific Inc., Waltham, MA) read the sample absorbance at the 660.6, 642.2, and 470 nm wavelengths to determine chlorophyll a, b, and carotenoids, respectively, in each leaf extract sample. Absorbance values were calculated using equations by Lichtenthaler [51] for extraction using diethyl ether. Calculations for the total concentrations of the pigments in cucumber leaf tissue:

Chlorophyll a (C_a_): 10.05A_660.6_ − 0.97A_642.2_

Chlorophyll b (C_b_): 16.36A_642.2_ − 2.43A_660.6_

Chlorophyll a + b (C_a+b_): 7.62A_660.6_ + 15.39A_642.2_

Carotenoids:1000A_470_ − 1.43C_a_ − 35.87C_b_/209

*Proline Analysis.* Proline content was determined by methods established by Bates et al. [52]. Freeze-dried cucumber leaf tissues were weighed to 0.5 g and placed in a stone mortar on ice and homogenized in 10 mL of 3% sulfosalicylic acid. The homogenate was filtered through Whatman #1 filter paper and collected in flasks. Two mL of the filtrate was then reacted with 2 mL of ninhydrin reagent, and 2 mL of glacial acetic acid in a microtube and placed for one hour in a 100 °C water bath. The reaction was stopped by placing samples in an ice bath. Then 2 mL of toluene was added to each reaction mixture and mixed vigorously. The reaction separated the solution, and the upper two mL containing toluene were transferred to glass cuvettes, and the absorbance was read at 520 nm using a Genesys 10S UV/Vis spectrophotometer (Thermo Fisher Scientific Inc., Waltham, MA, USA). Toluene was used as a blank, and calibration curves were determined before each use of the spectrophotometer. The equation for calculation of proline is: (µg proline/mL × mL toluene)/115.5 µg/µmol]/((g sample)/5) = µmoles proline/g of fresh weight material.

*Soluble Sugar Analysis.* Soluble sugar analysis was conducted according to Barickman et al. [53]. Briefly, lettuce leaf samples were ground in a bullet grinder for homogenous sub-samples. A 2.0 g sub-sample was extracted in a 15 mL test tube by adding 2 mL of reverse osmosis water, vortexing, and shaking for 15 min at 200 rpm. Samples were then centrifuged at 4000 rpm for 10 min, and 1.0 mL of the supernatant was transferred into a new 15 mL test tube. After the transfer, 1.4 mL of acetonitrile was added, and tubes were mixed by inversion and kept at room temperature for 30 min. Samples were then centrifuged at 4000 rpm for 10 min, and 1.0 mL of the supernatant was transferred into a new 15 mL tube and placed into a dry-bath until complete evaporation. Once dried, samples were dissolved in 0.5 mL of 75% acetonitrile and 25% RO water. Samples were then put through a 0.2-µm syringe filter and collected in a 2 mL HPLC vial for analysis. Separation parameters and sugar quantification were carried out with authentic standards using an Agilent 1100 series HPLC with a refractive index detector (Agilent Technologies, Wilmington, DE, USA). Chromatographic separations were achieved using a 250 × 4.6 mm i.d., 5 μm analytical scale NH_2_ (amino) carbohydrate C_18_ reverse-phase column (Agilent Technologies), which allowed for effective separation of chemically similar sugar compounds. The column was equipped with a Zorbax NH_2_ 4.6 × 12.5 mm i.d. guard cartridge and holder (Agilent Technologies) and was maintained at 30 °C using a thermostatted column compartment. All separations were achieved isocratically using a binary mobile phase of 75% acetonitrile and 25% RO (reverse osmosis) water (v/v). The flow rate was 1.0 mL min^−1^, with a run time of 15 min, followed by a 2 min equilibration prior to the next injection. Eluted compounds from a 10-µL injection loop were detected in positive detection mode, and data were collected, recorded, and integrated using ChemStation Software (Agilent Technologies). Peak assignment for individual sugars was performed by comparing retention times from the refractive index detector using external standards of fructose and glucose (Sigma–Aldrich, St. Louis, MO, USA).

*Statistical Analysis.* Statistical analysis of the data was performed using SAS (version 9.4; SAS Institute, Cary, NC, USA). Data were analyzed using the PROC GLIMMIXED analysis of variance (ANOVA) followed by mean separation. The experimental design was a randomized complete block with two treatments, four blocks, and two replications. The fixed effect for the experiment consisted of the two water treatments, while blocks and replications were analyzed as random effects. The standard errors were based on the pooled error term from the ANOVA table. Duncan’s multiple range test (*p* ≤ 0.05) was used to differentiate between waterlogging and non-waterlogging classifications when *F* values were significant for main effects. Model-based values were reported rather than the unequal standard error from a data-based calculation because pooled errors reflect the statistical testing being done. Regression analysis was used to study changes associated with the treatments and physiological measurements. Diagnostic tests were conducted to ensure that treatment variances were statistically equal before pooling.

## 4. Conclusions

The current study reveals the physiological difference between waterlogged and non-waterlogged cucumber plants. The results demonstrated that waterlogging of sensitive cucumber plants early in development stages causes a reduction in numerous physiological responses. Maintenance of normal leaf respiration by cucumber plant under waterlogging stress may be associated with their adaptability, such as the development of adventitious roots. However, waterlogging at an early vegetative stage caused a severe reduction in the physiological performance by decreasing *A*, and C_i_. However, g_s_ and *E* was reduced in the first initial days of waterlogging treatment initiation, while recovering from having no differences by days four through 10. Cucumber Chl, proline, and sugar content were also significantly affected by a 10-day waterlogging treatment. While photosynthetic efficiency was reduced by waterlogging, some initial tolerance mechanisms were also seen. The relationships between the reduction of *A*, biomass, and photosynthetic pigments were evident in waterlogged compared to non-waterlogged treated plants. Waterlogging caused an overall negative effect on young cucumber plants as they had a decreased plant height, leaf area, and biomass accumulation. However, these results show that stress responses may impart a partial tolerance to waterlogging events. These responses could potentially allow for recovery after waterlogging occurs. Thus, a deep and broad understanding of early developmental waterlogging stress is necessary to recognize how plants respond to these devastating effects.

## Figures and Tables

**Figure 1 plants-08-00160-f001:**
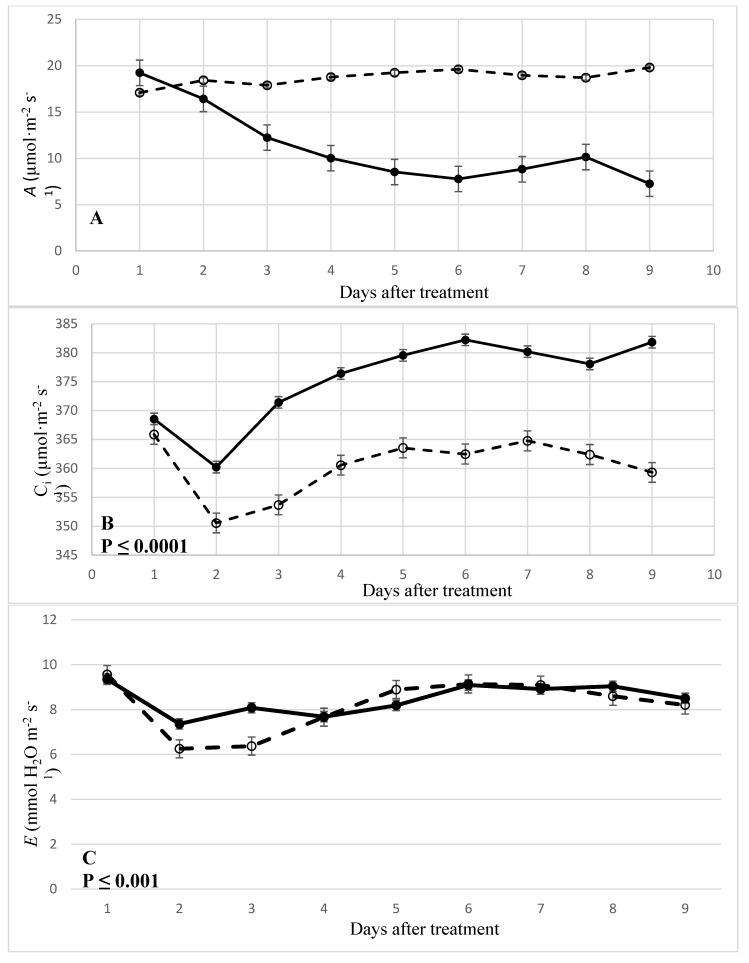
The effect of waterlogging on (**A**) net photosynthesis (*A*), (**B**) internal CO_2_ (C_i_), (**C**) transpiration (*E*), and (**D**) stomatal conductance (g_s_) of cucumber plants in an early developmental stage of growth. Water was kept 10 cm above the container to simulate waterlogging stress. Data are from 10-day treatments of waterlogging (•) or non-waterlogging (○) treatments.

**Figure 2 plants-08-00160-f002:**
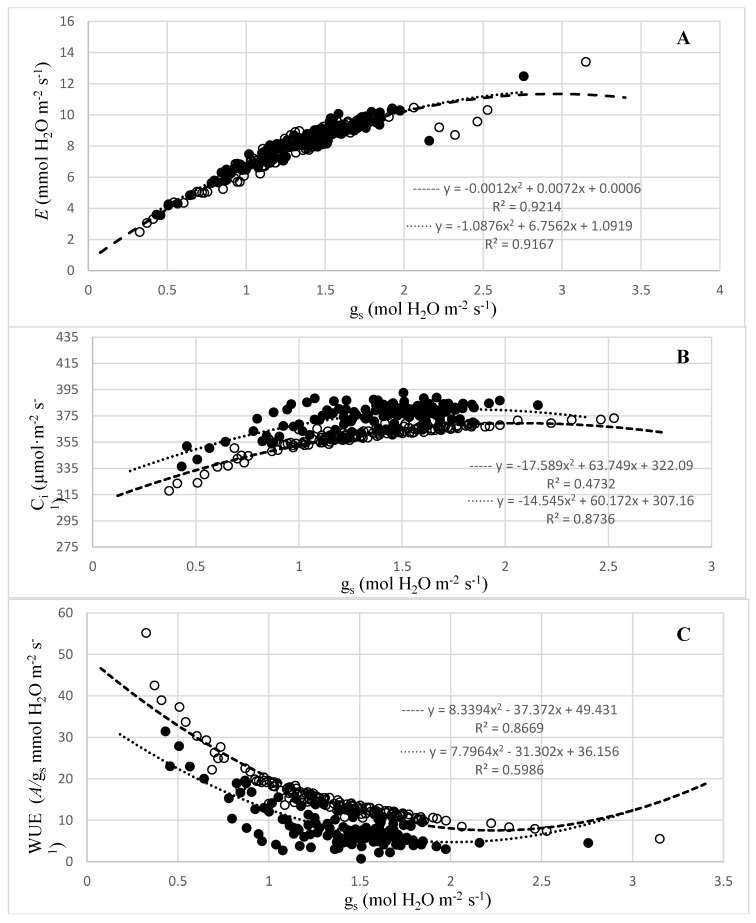
Relationship between stomatal conductance (g_s_) and (**A**) transpiration (*E*), (**B**) internal CO_2_ (C_i_), and (**C**) water use efficiency (WUE) in cucumber. Data is from 10-day treatments of waterlogging (•) or non-waterlogging (○) treatments (*p* ≤ 0.0001).

**Figure 3 plants-08-00160-f003:**
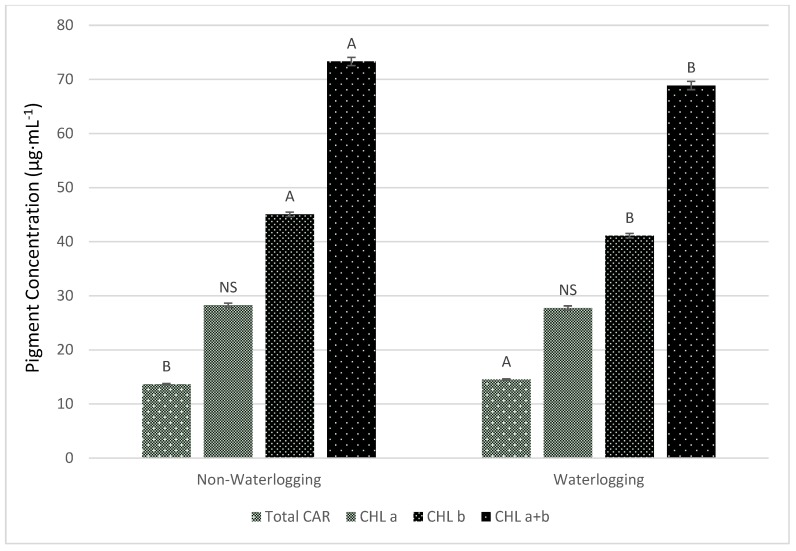
Total carotenoid (CAR), total chlorophyll (Chl a + b), chlorophyll a (Chl a), and chlorophyll b (Chl b) concentrations of cucumber plants under waterlogged or non-waterlogged conditions for 10 days. Plants were harvested and analyzed for CAR and Chl at the end of 10 days.

**Figure 4 plants-08-00160-f004:**
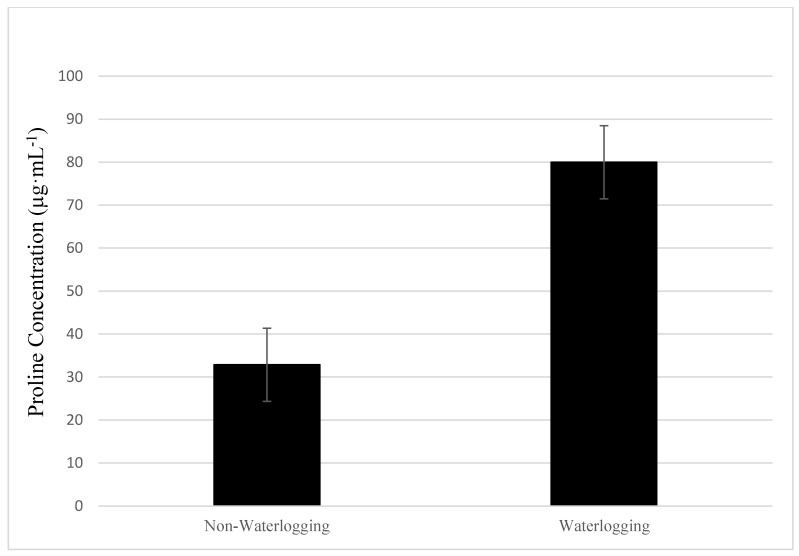
Total proline concentrations of cucumber plants under waterlogged or non-waterlogged conditions for 10 days. Plants were harvested and analyzed for proline at the end of 10 days.

**Figure 5 plants-08-00160-f005:**
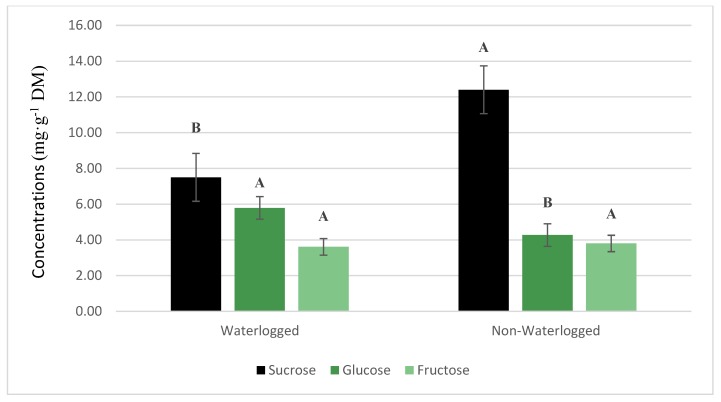
Soluble sugar concentrations in cucumber leaf samples taken from plants that were waterlogged or not waterlogged for 10 days. Plants were harvested and analyzed for soluble sugars at the end of 10 days.

**Table 1 plants-08-00160-t001:** The effects of a 10-day waterlogging or non-waterlogging treatment on cucumber plants in an early developmental stage. Plants were harvested at the end of 10 days.

Treatment ^z^	Ht (cm)	Leaf Number	Leaf Area (cm^2^)	Fresh Mass (g)	Dry Mass (g)	DM:FM	Water (%)
Waterlogging	15.13 ^b^	7.09 ^b^	411.28 ^b^	17.14 ^b^	2.68 ^b^	0.16 ^a^	84.36
Non-Waterlogging	17.25 ^a^	7.97 ^a^	650.77 ^a^	26.45 ^a^	3.62 ^a^	0.14 ^b^	86.31
*p-Value ^b^*	*	*	***	***	**	*	

^z^ Standard error of the mean: Ht = 0.66, Leaf number = 0.25, Leaf area = 23.69, Fresh. mass = 1.05, Dry mass = 0.14, DM:FM = 0.01. ^b^ *, **, *** indicate significance at *p* ≤ 0.05, 0.01, 0.001, respectively. Different letters indicate significant differences between treatments (*p* ≤ 0.05) according to Duncan’s multiple range test.

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
