# Peer review of "Waterlogging Causes Early Modification in the Physiological Performance, Carotenoids, Chlorophylls, Proline, and Soluble Sugars of Cucumber Plants"

_plants, 2019, doi:10.3390/plants8060160_

Round 1

Reviewer 1 Report

This is a quality presentation of a well conceived study.  The fact that is disputes prior root hydraulic conductance/ stomatal closure responses is not particularly troubling, and handled well in discussion.

There are some small language changes I would suggest, delineated by line number below.

47                           adventitious

77-75                     reads like introduction, not results and discussion

78-80                     what results are you referring to?

100                        if DM/FM were expressed as water content, then it would be a direct relationship

107                         why not show the data?

112                         awkward wording

239-241                awkward wording

361                         “together” is not needed

363                         early in development

Author Response

Reviewer 1:

Line 46: corrected spelling to adventitious

Lines 75-77: Thank you for the comment. Even though the first paragraph reads like an introduction, the authors feel that transitioning into the Results and Discussion section needs to have a flow that connects the sections together. Thus, the authors would like to leave the paragraph as transitionary to connect the sections together.

Lines 78-80: Thank you for pointing out the referral to results in these lines. The author has re-worded this sentence to have a broader context to the overall points of the manuscript. Thus, the wording “results” are replaced.

Line 100: The authors are not sure if the reviewer is stating that we should add that there is a direct relationship between the ratio of DM:FM and water content.

Line 107: Thank you for your comment. However, the authors did not show the data because the non-waterlogged cucumber plants did not produce any adventitious roots. The figure/table would show no bar graph or just a 0.0 in the table. Therefore, we choose not to show the data.

Line 112: This is a summary sentence that states in general, previous research is in agreement that waterlogged/flooded plants have decreased photosynthesis. We did not want to be redundant with how we start off sentences.

Lines 239-241: Yes, thank you for pointing out this awkward sentence. The authors have corrected this sentence to better articulate the statement.

Line 361: Thank you for your comment. The authors have changed the beginning of the sentence to “The current study….”

Line 363: The authors have corrected the spelling and grammar of the sentence.

Reviewer 2 Report

This is an interesting study as waterlogging is a common stress environmental factor affecting large agricultural areas of the word.

However, the manuscript should be revised and specially the introduction and discussion rewritten in a clearer, concise manner, avoiding the use of redundant sentences that make the manuscript difficult to understand.

The discussion is poor and could be improved by analyzing the relationships among the different plant growth and physiological parameters studied to conclude by stating their relative importance in the growth of waterlogged cucumber seedlings 

My comments are as follows,

A brief sentence stating the main conclusions of this study should be included in the abstract section.

The aims of this study should more orderly stated in the Introduction. What are effects of waterlogging? and what are plant responses to waterlogging?

Which growth parameter (table 1) was the most affected by waterlogging? Were all similarly inhibited? Could the decrease in leaf area in waterlogged plants explain the difference in plant dry mass between treatments?

Could Figure 1 show a possible transplanting effect on gas exchange parameters as both, waterlogged and non-waterlogged plants showed decreased stomatal closure.?

From day 3, the non-stomatal components limited CO2 assimilation, which is also visualize in Figure 2B, where, for a given gs, waterlogged plants showed higher Ci than the non-waterlogged ones. Could that be related to the reduced photosynthetic pigment concentration found in the waterlogged plants? These types of relationships should be introduced and discussed in the manuscript.

Also, in table 1, data on the plant water status could be provided, for example plant water content (FM-DM/DM) and its relationship with other measured parameters discussed.

It is interesting that CO2 assimilation showed much higher inhibition (65%) (Fig. 1) than plant growth (25%) (Fig.2), could the authors elaborate more in the discussion on the relationship between the different parameters analyzed? It would greatly help them to improve the conclusion section.

Figures should be downsized. 

Y axis in Figure 5 states “Dry weight”, please correct.

Author Response

This is an interesting study as waterlogging is a common stress environmental factor affecting large agricultural areas of the word.

However, the manuscript should be revised and specially the introduction and discussion rewritten in a clearer, concise manner, avoiding the use of redundant sentences that make the manuscript difficult to understand.

The discussion is poor and could be improved by analyzing the relationships among the different plant growth and physiological parameters studied to conclude by stating their relative importance in the growth of waterlogged cucumber seedlings 

My comments are as follows,

A brief sentence stating the main conclusions of this study should be included in the abstract section.

Thank you for your comments. The authors have added a brief sentence from the conclusion to the abstract.

The aims of this study should more orderly stated in the Introduction. What are effects of waterlogging? and what are plant responses to waterlogging?

Thank you for your suggestions. The authors have included on Lines 65-66 the physiological effects and on Line 70, the authors have included the same parameters in objective 2.

Which growth parameter (table 1) was the most affected by waterlogging? Were all similarly inhibited?

The parameters most affected by waterlogging were leaf area and fresh mass. The statistics indicated that these two parameters were effected the most by waterlogging. The authors have included a sentences stating this fact in Lines 103-105.

Could the decrease in leaf area in waterlogged plants explain the difference in plant dry mass between treatments? Thank you for pointing out that the differences in dry mass between the treatments were caused by decreases in leaf area. The author has included a sentence to articulate that in Lines 105-106.

Could Figure 1 show a possible transplanting effect on gas exchange parameters as both, waterlogged and non-waterlogged plants showed decreased stomatal closure.? Thank you for the suggestion. The authors have added a sentence on Lines 161-163.

From day 3, the non-stomatal components limited CO2 assimilation, which is also visualize in Figure 2B, where, for a given gs, waterlogged plants showed higher Ci than the non-waterlogged ones. Could that be related to the reduced photosynthetic pigment concentration found in the waterlogged plants? These types of relationships should be introduced and discussed in the manuscript. Thank you for pointing out this relationship. The authors have included a couple of sentences regarding the relationship in Lines 224-228.

Also, in table 1, data on the plant water status could be provided, for example plant water content (FM-DM/DM) and its relationship with other measured parameters discussed.

The authors have added the water content (%) in table 1. The authors added sentences on Lines 110-113; 199-200

It is interesting that CO2 assimilation showed much higher inhibition (65%) (Fig. 1) than plant growth (25%) (Fig.2), could the authors elaborate more in the discussion on the relationship between the different parameters analyzed? It would greatly help them to improve the conclusion section. Thank you for pointing out these relationships between these two parameters. The authors have included discussion on Lines147-157.

Figures should be downsized. 

The authors suggest that the editors and publishers make this decision regarding the size and placement of the figures.

Y axis in Figure 5 states “Dry weight”, please correct.

Thank you for catching the mistake. Figure 5 is corrected from DW to concentrations

Round 2

Reviewer 2 Report

In my opinion, after the changes introduced by the authors, the paper can be accepted on Plants